# Children’s Iodine Intake from Dairy Products and Related Factors: A Cross-Sectional Study in Two Provinces of China

**DOI:** 10.3390/nu16132104

**Published:** 2024-07-02

**Authors:** Ying Zhang, Haiyan Wang, Wei Ma, Xiuwei Li, Jianqiang Wang, Jinpeng Wang, Jing Xu

**Affiliations:** Key Laboratory of Public Nutrition and Health, National Health Commission of the Peoples’ Republic of China, National Institute for Nutrition and Health, Chinese Center for Disease Control and Prevention, No 155 Changbai Road, Changping District, Beijing 102206, China; zhangying@ninh.chinacdc.cn (Y.Z.); wanghy@ninh.chinacdc.cn (H.W.); mawei@ninh.chinacdc.cn (W.M.); lixw@ninh.chinacdc.cn (X.L.); wangjq@ninh.chinacdc.cn (J.W.); lpl6661010@163.com (J.W.)

**Keywords:** iodine intake, iodine content, milk, dairy products, children

## Abstract

Dairy products are a significant source of iodine, and their contribution to iodine intake must be evaluated regularly. However, there is a lack of data on iodine intake from dairy products in China. Through a cross-sectional study, we determined the iodine content of dairy products in the Chinese diet and estimated iodine intake among Chinese children. Intake records for 30 consecutive days were used to investigate the consumption of dairy products by 2009 children from Yunnan and Liaoning Provinces. The iodine contents of 266 dairy products with high intake frequency were determined using inductively coupled plasma–mass spectrometry (ICP-MS). We then calculated the iodine intake and contribution of dairy products and explored the related factors of dairy iodine intake through a generalized linear mixed model. Ultra-high-temperature (UHT) sterilized milk accounted for 78.7% of the total dairy products, with an iodine content of 23.0 μg/100 g. The dairy product intake rate of children in China was 83.6%, with an average daily intake of 143.1 g. The median iodine intake from milk and dairy was 26.8 μg/d, 41.5% of the estimated average recommendation (EAR) for younger children and 31.8% of the EAR for older children. The daily milk iodine intake of children in Yunnan Province was 9.448 μg/day lower than that of children in Liaoning Province (*p* < 0.001), and the daily iodine intake of children in rural areas was 17.958 μg/day lower than that of children in urban areas (*p* < 0.001). Chinese dairy products were rich in iodine, and the content of iodine was intermediate to that reported in Europe and the USA. However, children’s daily intake of milk iodine was lower than that of children in other developed countries due to the lower daily intake of dairy products, especially those in rural areas.

## 1. Introduction

Iodine, a crucial element for the production of thyroxine and triiodothyronine, plays a pivotal role in normal neurodevelopment and growth [1]. Its deficiency in children can lead to impaired mental functions and delayed physical development [2]. Conversely, an excess of iodine can also be detrimental, causing conditions such as goiter, hyperthyrotropinemia, and autoimmune thyroid disease [3]. The tolerable upper intake level (UL) of iodine for children aged 7–11 years is 250 μg/d, significantly lower than the 600 μg/d for adults [4]. Given the narrow range of safe iodine intake, supplementation may pose a risk of iodine excess. Therefore, it is crucial to monitor the sources of dietary iodine intake and collect accurate intake information on children to maintain iodine sufficiency.

In recent decades, dairy products such as milk, in addition to iodized salt, have emerged as the primary source of dietary iodine in many developed countries. They are estimated to contribute 25–70% of daily iodine intake [5]. In France, 40% of dietary iodine for children and adolescents is derived from dairy products [6]. Milk accounts for approximately 35% of the daily dietary iodine intake of preschool children in Flanders [7], 38% of the iodine intake of children aged 6–12 years in Germany [8], 50% of the iodine intake of children aged 4–13 years in Norway [9], and 51–76% of the iodine intake of children and adolescents in the USA [10]. However, unlike in developed countries, there is a dearth of data on the iodine intake from dairy products in China. This could be attributed to two main reasons: Firstly, it is widely believed that the consumption of dairy products in China is relatively low, and the contribution of dairy iodine intake to the daily iodine requirement is insignificant; secondly, there is a lack of primary data on the iodine content of different dairy products in our country, which hampers the calculation of the iodine content ingested from dairy products.

Although people tried to drink milk as early as 3000 years ago during the Shang Dynasty, it was in the early 20th century that milk was formally associated with national health. In 1949, the average annual milk intake per person was only 400 g or 1.1 g per day. At that time, milk consumption mainly appeared in Beijing and a few big coastal cities, and only a few well-off families and some special people could enjoy this uncommon nutritious product. After China’s reform and opening up, milk consumption began to spread and proliferate. The latest edition of the Dietary Guidelines for Chinese Residents recommends that each person drink 300 g of milk per day or an equivalent amount of dairy products [11], and milk has become an important part of the Chinese daily diet. Since 2000, China’s government has vigorously promoted the Student Milk Drinking Program to improve students’ physical fitness and nutritional health [12]. With the increase in the rate and volume of milk consumption by children, it is necessary to estimate the contribution of milk and dairy products to iodine intake in Chinese children. This requires information on the iodine content of dairy products and their consumption level. However, data on iodine content for Chinese dairy products are scant and must be updated [13]. Therefore, the study objectives included the following: (1) determining the current iodine content of Chinese dairy products, (2) investigating the rate and intake of dairy products among children, (3) estimating their contribution to children’s iodine intake, and (4) exploring factors influencing milk iodine intake.

## 2. Materials and Methods

### 2.1. Survey Areas and Participants

This cross-sectional study was conducted in China from 2022 to 2023. We selected Liaoning Province as the representative of the northern region and Yunnan Province as the representative of the southern region. One city was randomly selected from each province in the five directions: east, west, south, north, and center. One district was randomly selected from the city to represent the urban area, and one county was selected to represent the rural area. Each of the districts or counties selected was divided into five sampling areas according to east, west, south, north, and center, with one street randomly selected in each of the districts and one village randomly selected in each of the counties. Twenty children were randomly selected in each of the streets or villages. The inclusion criteria comprised the following: (1) children aged 7–14 y; (2) an equal number of boys and girls; (3) healthy children without malnutrition, or cardiovascular or thyroid diseases; and (4) children who consented to participate in all study aspects. 

### 2.2. Questionnaire

Before the survey, a paper recording form was given to each child and guardian to record their intake of dairy products for the next 30 consecutive days, including the date, category, name, and brand of dairy products consumed and the amount consumed per serving. To ensure the authenticity and accuracy of the survey data, each respondent was given a clean plastic bag labeled with the child’s name and contact information before the survey. The children were instructed to collect the packaging in a plastic bag after each consumption of dairy products. At the end of the study, all the packages of dairy products were handed over to the staff along with the record form so that we could easily verify the accuracy of the information filled in. The participants provided demographic data, including age, gender, home address, and ethnicity.

### 2.3. Dairy Product Selection and Sampling

We analyzed the dairy intake data and identified seven categories of dairy products: pasteurized milk, UHT milk, yogurt, milk powder, cheese, cream or butter, and others. Each province selected items for analysis of their iodine content. In total, 266 products were selected for purchase and analysis.

### 2.4. Analysis of Dairy Iodine Content

The iodine in the dairy product was extracted by tetramethyl ammonium hydroxide (TMAH) and determined by inductively coupled plasma–mass spectrometry (ICP-MS). The solid dairy products were pulverized by a high-speed pulverizer and stirred until homogeneous, and the powdered or liquid dairy products were shaken well. A test sample of 0.2 g~1 g (accurate to 0.001 g) was added to 5 mL of extraction solution (5% TMAH). Extraction was carried out in a constant-temperature water bath shaker at 85 °C for three hours. After cooling, the sample solution was diluted to a final volume of 50 mL with deionized water and centrifuged at more than 3000 r/min for 10 min. The upper layer of the clear solution was filtered through a 0.45 μm filtration membrane. The iodine standard solution and 500 μg/L rhenium internal standard were injected into the ICP-MS to determine the signal response value of iodine and internal standard elements. The standard curve was plotted with the concentration of iodine as the horizontal coordinate and the ratio of iodine to the response signal value of the selected internal standard element as the vertical coordinate. The blank and sample solution were injected into the ICP-MS, the signal response values of iodine and the selected internal standard element were measured, and the concentration of iodine in the solution was measured by the standard curve [14]. Iodine content is reported as μg/100 g.

### 2.5. Calculation of Dairy Products and Definition of Variables

Records of 30 consecutive days were used to investigate the categories, frequency, and weight of dairy products consumed by children, which led to the average 30-day intake of each category of dairy products for each child. In this study, dairy product mass was converted to milk mass according to protein content and was harmonized by multiplying the original weight by the corresponding conversion coefficients [15]. Individual dietary iodine intake from dairy products was calculated for each participant by multiplying the amount of each product consumed by the iodine content of that product.

Dairy consumers were those who consumed more than 0 g of dairy products per day, and the dairy intake rate (DIR) was the number of dairy consumers divided by the total number of children. We described the distribution and median dairy iodine intake for the children and stratified by gender, age, province, area, and ethnicity. We classified children <12 years as younger children and children ≥12 years as older.

### 2.6. Statistical Analysis

All data analyses were performed using the Statistical Analysis System version 9.4 for Windows (SAS, Institute Inc., Cary, NC, USA). The distribution of continuous variables was tested using the Kolmogorov–Smirnov test. The normally distributed data were expressed as means and standard deviations, and non-normally distributed data as medians and interquartile ranges. Categorical variables were expressed as numbers and percentages. In univariate analysis, *p*-values were derived from the Mann–Whitney nonparametric test to determine differences between groups with continuous skewed data for two-group comparison. The proportional compositions of the two groups were compared using Chi-square or Fisher’s exact tests. In multivariate analysis, individual-level variables were the gender of the child, age group, and ethnicity, and community-level variables were residence and province. Since the outcome variable, milk iodine intake, was continuous data with a skewed distribution, and the nature of the data was hierarchical, a two-level generalized linear mixed model (GMML) was fitted using an identity link to identify the independent effects of individual-level and community-level (cluster) variables on milk iodine intake. Four distinct models were constructed. The null model did not contain any independent variables. Model I represented a multivariable adjustment focused on individual-level factors, while Model II was adjusted for community-level variables. In Model III, the outcome was enhanced with potential variables from both the individual and the community level. The final model chosen featured the lowest Akaike Information Criterion (AIC) and Bayesian Information Criterion (BIC). For all comparisons, *p* < 0.05 was considered statistically significant.

## 3. Results

### 3.1. Characteristics of Participants

In this study, 1010 children were surveyed in Liaoning Province, 999 children were surveyed in Yunnan Province, and a total of 2009 respondents were included for analysis. Boys were 51.6% and girls 48.4%, with an average age of 11.0 ± 0.9 years. Of these children, 49.9% came from urban areas and 50.1% from rural areas. Han Chinese children accounted for 66.1%, and children from other ethnic minorities accounted for 33.9% (Table 1).

### 3.2. Dairy Product Intake Rate and Consumption

The DIR was 83.6%, with girls slightly higher than boys, younger children slightly higher than older children, and Liaoning Province higher than Yunnan Province, but the differences were not statistically significant (χ^2^ = 1.97, *p* = 0.16; χ^2^ = 0.15, *p* = 0.70; χ^2^ = 2.80, *p* = 0.09). The DIR among urban children was 92.3%, which was significantly higher than that of rural children (χ^2^ = 111.69, *p* < 0. 001) (Table 1).

The Dietary Guidelines for Chinese Residents (2022 edition) recommends that children consume more than 300 g of milk daily. However, only 5.1% of the children in this study consumed more than 300 g of milk per day. Only 1.2% of the older children consumed more than 300 g of milk, significantly lower than the younger children (χ^2^ = 9.76, *p* = 0.002). Among urban children, 9.2% had a milk intake ≥300 g, which was significantly higher than rural children (χ^2^ = 67.40, *p* < 0.001). There were no statistically significant differences in the proportion of children with a milk intake greater than 300 g by gender, province, or ethnicity (χ^2^ = 0.14, *p* = 0.71; χ^2^ = 0.03, *p* = 0.87; χ^2^ = 2.90, *p* = 0.09) (Table 1).

The children’s average milk intake was 143.1 g/day. The daily milk intake of children in Liaoning Province was 57.6 g more than that of children in Yunnan Province, which was a significant difference (Z = −6.76, *p* < 0.001). The daily milk intake of children in urban areas was 1.9 times higher than that of those in rural areas, which was a significant difference (Z = 13.23, *p* < 0.001). There was no statistically significant difference in milk intake between genders, age groups, or ethnic groups (*p* > 0.05) (Table 1).

### 3.3. Iodine Content, Intake of Dairy Products, and Milk Iodine Intake by Category

Table 2 shows the iodine content, children’s average consumption, and iodine intake by dairy product category. Milk powder had the highest iodine content of all dairy products, 125.5 μg/100 g. UHT milk had an iodine content of 23.0 μg/100 g. The intake of UHT milk accounted for 78.7% of the total dairy products, which is a significant part of the dairy products, while yogurt intake accounted for 9.8%, pasteurized milk accounted for 6.6%, and powdered milk accounted for 1.3%. The above data show that UHT milk is the leading dairy food for children in China. On average, drinking UHT milk provided 79.0% of iodine intake from dairy products. Yogurt and pasteurized milk provided 11.5% and 7.1% of the intake of iodine.

### 3.4. Milk Iodine Intake and Related Factors

Iodine intake from dairy products in the children was positively skewed, with a median of 26.8 µg/day. In the univariate analysis, the distribution of iodine intake from dairy products differed between province, residence, and ethnicity (Table 3). The median iodine intake among children from Yunnan Province was significantly lower than Liaoning Province’s median (Z = −8.47, *p* < 0.001). Urban children had significantly higher iodine intake from milk and dairy than rural children (Z = 12.30, *p* < 0.001). The median iodine intake was 29.7 μg/day for Han children vs. 23.1 μg/day for minority children (Z = −3.70, *p* < 0.001). Iodine intake distributions did not differ significantly by gender or age group (*p* > 0.05).

In multivariate analysis, we compared the three models. Model III, i.e., the model adjusted for individual and community-level factors, had a smaller AIC and BIC (Table 4). The results from the final multivariable Model III showed that the province, place of residence, and age of the child were significantly associated with iodine intake. The daily milk iodine intake of children in Yunnan Province was 9.448 μg/day lower than that of children in Liaoning Province (*p* < 0.001), and the daily iodine intake of children in rural areas was 17.958 μg/day lower than that of children in urban areas (*p* < 0.001) (Table 5).

### 3.5. Distribution of Milk Iodine Intake and the Proportion of the EAR, RNI, and UL for Iodine

In our study, we found that 87.1% of children’s milk iodine intake was below the daily EAR. Only 5.7% of younger and 0.8% of older children consumed iodine higher than the RNI. Without considering other iodine intake, the daily milk iodine intake of younger and older children accounted for 41.5% and 31.8% of the EAR, respectively. The most common milk package in China is 250 mL per serving, and the recommended daily milk intake for children is 300 mL. It is worth noting that milk, particularly UHT milk, is crucial for providing iodine to children. Using the average iodine concentration of all UHT milk, we calculated that one serving provides 88.5% of the EAR of iodine for younger and 71.9% for older children. The recommended milk intake, therefore, provides a substantial 106.2% of the EAR for younger children and 86.3% of the EAR for older children (Table 6).

## 4. Discussion

This study tested 266 Chinese dairy products with high consumption rates for iodine content. The study investigated 2009 children’s dairy product intake, milk iodine intake, and the contribution of dairy products to iodine RNI. It revealed significant differences in milk iodine intake between factors such as province and residence.

The average iodine content of UHT milk, most frequently consumed by children, was 23.0 μg/100 g, and the average iodine content of yogurt was 20.4 μg/100 g, which was significantly higher than the iodine content of milk (1.9 μg/100 g) and yogurt (0.9 μg/100 g) reported before 2010 [13]. Dutch cow milk was reported to have lower iodine concentrations than Chinese milk (15.9 vs. 23.0 μg/100 g) [16]. Data from Norway showed slightly lower iodine content in comparison to China for milk (20.0 vs. 23.0 μg/100 g) and yogurt (18.0 vs. 20.4 μg/100 g) [17]. However, we found lower iodine levels in milk from China than the values reported for the UK (23.0 vs. 31.4 μg/100 g) [18] and the USA (23.0 vs. 35.4 μg/100 g) [19]. Iodine contents in dairy products vary widely within and between countries, depending on the iodine content of water, animal feed, season, and use of iodine as a disinfectant, so it is necessary to monitor the iodine content of dairy products regularly and update the data in a timely manner. Twenty percent or more of the daily value (DV) of a nutrient per serving is considered high. Therefore, dairy products in China are an excellent source of dietary iodine.

With the rapid development of China’s economy and the increase in residents’ concern for nutrition, the intake rate and number of dairy products for children in China have been increasing. The survey of milk consumption of children aged 7–17 years in nine provinces showed that the milk consumption rate was only 3.0% in 1991 and 14.1% in 2006, with a daily milk consumption of 3.9 ± 31.9 g/d in 1991 and 26.7 ± 85.0 g/d in 2006 [20]. Data from the Nutrition and Health Monitoring of Chinese Residents showed that in 2012, the milk consumption rate of children aged 6–17 in Liaoning Province was 35.1%, and the average milk consumption was 49.2 g/d [21]. Data obtained from the Nutrition Improvement Program for Rural Compulsory Education Students (NIPRCES) showed that the milk-drinking rate of children aged 8–15 was 80.6% in 2019 and 88.4% in 2021 [22], similar to the results of our study. Especially since the outbreak of COVID-19, there has been a significant increase in the emphasis on nutritional health and immunity enhancement, with a more diversified dairy intake and an increase in the number of people drinking milk and the amount consumed.

Despite the average dairy intake of 143.1 g/day, children had a low dairy consumption compared to other countries. In Germany [23], the daily intake of milk products for children aged 9–13 years is 391 g for boys and 304 g for girls. In France [24], the average milk, cheese, and chilled dairy product intake for children aged 3–17 years were 177.2, 18.8, and 76.0 g/day, respectively. Our children’s dairy intake is much lower than that of Western developed countries, probably due to differences in dietary habits. However, our children’s dairy intake is also lower than that of Singapore [25], which has a similar diet to our country, reflecting that there is still much work to do to increase our children’s dairy intake.

While our laboratory test shows that dairy products in 2022–2023 are potentially good sources of iodine for the Chinese diet, the survey suggested that consumption of dairy products did not ensure adequate iodine intake among Chinese children. Studies in other developed countries revealed higher iodine intake from dairy products. The mean intakes of iodine from milk, cheese, and dairy products among 1722 children and adolescents in Norway were higher than what we found among Chinese children (56 μg/day vs. 26.8 μg/day) [26]. The average daily intake of dairy iodine for children aged six years and ten years in the USA was 158.8 μg/day and 140.5 μg/day, respectively, which was 5–6 times higher than that of our children [10]. If other dietary iodine intakes are not taken into account, the daily milk iodine intake in this study would only account for 41.5% of the EAR for younger children and 31.8% of the EAR for older children. In addition, 87.1% of the children would have insufficient iodine intake, indicating the need for children to consume other iodine-rich foods. Consumption of 300 mL milk daily, recommended by the Chinese Nutrition Society, would provide 106.2% of EAR for younger children and 86.3% of RNI for older children. This suggests that increased consumption of dairy products could reduce the risk of iodine deficiency among Chinese children.

The generalized linear mixed model confirms the significant association of milk iodine intake with the province and residence. The median iodine intake among children from Yunnan Province was significantly lower than that of those from Liaoning Province. This could be due to several factors, including differences in dietary habits, variations in iodine content in milk, or varying levels of dairy consumption awareness. The urban–rural divide in iodine intake from dairy products is another striking finding. Urban children had significantly higher iodine intake than their rural counterparts, indicating a potential health disparity. This could be attributed to greater access to various dairy products, higher income levels, and better education on nutrition in urban areas. A similar trend was observed by Fan et al. [15], who reported that as per capita income status and education levels increased, the level of dairy intake of the population also increased. Gender and age group did not significantly affect iodine intake distributions in this study. This suggests that the observed differences are not due to biological differences between genders or age-related changes in dietary habits. However, it is essential to maintain vigilance regarding these factors since other studies have reported variations in dairy intake based on gender and age [15].

The results of this study have significant implications for the development of public health policies aimed at improving childhood nutrition in China. At the national level, our findings emphasize the need for policies ensuring equal access to nutritious foods such as milk in both urban and rural communities. Most children in China do not meet the nationally recommended standard of 300 g of milk per day, putting them at risk of iodine deficiency if they do not consume other iodine-containing foods or iodized salt. The Chinese Government should continue to promote children’s consumption of 300–500 g of dairy products daily and adhere to the policy of iodizing salt in iodine-deficient areas. Moreover, national campaigns to raise awareness about the importance of iodine and other essential nutrients could be strengthened to encourage healthier eating habits. At the provincial and local levels, our research suggests a need for targeted interventions to address regional disparities in milk consumption. Local health departments could work with schools to provide nutrition education and ensure that school meal programs include a 300 g serving of dairy products. In terms of clinical practice, healthcare providers should be informed about the potential for iodine deficiency in children who do not consume milk or other iodine-rich foods. Pediatric guidelines may stress the importance of including iodine sources in children’s diets, and clinicians could play a crucial role in educating parents about the risks associated with insufficient iodine intake and strategies to tackle this issue.

This is the first large-scale study to determine iodine content in Chinese dairy products and investigate milk iodine intake among children. In addition, we used an intake record for 30 consecutive days to record children’s dairy intake, avoiding recall bias and non-representativeness due to 3-day dietary records. Our study has several limitations. First, we only investigated dairy intake and no other food intake, so we could not derive the contribution of dairy products to the daily total iodine intake, only the contribution of milk iodine intake to the RNI. Second, we investigated children’s milk iodine intake in only two provinces, which is not nationally representative. Next, we will conduct studies of milk iodine intake among crucial populations in multiple provinces, including pregnant women and lactating mothers. Finally, we did not consider the seasonal variability of iodine content in dairy products. The dairy products in the study were all produced in the same month. Next, we will monitor the iodine content of dairy products in different months.

## 5. Conclusions

In conclusion, Chinese dairy products were rich in iodine, and the content of iodine was intermediate to that reported in Europe and the USA. However, children’s daily intake of milk iodine was lower than that of children in other developed countries due to their lower daily intake of dairy products, especially those in rural areas. The current intake of dairy products for children does not provide enough iodine to meet the physiological needs of children, and adequate iodine needs to be obtained from other foods, such as iodized table salt.

## Figures and Tables

**Table 1 nutrients-16-02104-t001:** Dairy intake rate and amount in different subgroups of the population.

Variables	N	DIR (%)	Milk Intake (g) N		Milk Intake (g)
			<300	≥300	
Gender					
Boy	1037	82.5	982	55	150.0 (183.3)
Girl	972	84.8	924	48	133.3 (161.3)
Age					
<12 years	1748	83.7	1648	100 ^a^	143.0 (169.6)
≥12 years	261	82.8	258	3	143.8 (183.0)
Province					
Liaoning	1010	85.0	959	51	161.8 (158.0) ^a^
Yunnan	999	82.2	947	52	104.2 (175.6)
Residence					
Urban	1003	92.3 ^a^	911	92 ^a^	176.8 (136.3) ^a^
Rural	1006	74.9	995	11	92.9 (200.0)
Ethnic					
Han	1327	83.6	1251	76	147.3 (186.7)
Other	682	83.6	655	27	129.4 (165.4)
Total	2009	83.6	1906	103	143.1 (174.1)

DIR, dairy intake rate; N, number. ^a^ The differences between groups were statistically significant, *p* < 0.001.

**Table 2 nutrients-16-02104-t002:** Iodine content and iodine intake of different dairy product categories.

Category	N	Iodine Content (μg/100 g)	Consumption (g)			Iodine Intake (μg)		
			Mean ± SD	Median (IQR)	%	Mean ± SD	Median (IQR)	%
Pasteurized milk	27	20.5 (15.9)	9.6 ± 45.9	0.0 (0.0)	6.6	2.4 ± 11.8	0.0 (0.0)	7.1
UHT milk	117	23.0 (17.6)	113.8 ± 104.5	100.0 (200.0)	78.7	27.0 ± 28.9	19.5 (44.6)	79.0
Yogurt	85	20.4 (16.2)	14.2 ± 34.5	0.0 (0.0)	9.8	3.9 ± 10.2	0.0 (0.0)	11.5
Milk powder	14	125.5 (87.4)	1.9 ± 14.4	0.0 (0.0)	1.3	0.3 ± 3.4	0.0 (0.0)	1.0
Cheese	12	17.7 (11.1)	3.7 ± 43.7	0.0 (0.0)	2.6	0.0 ± 0.6	0.0 (0.0)	0.2
Cream or butter	7	4.0 (0.0)	0.3 ± 3.0	0.0 (0.0)	0.2	0.2 ± 2.4	0.0 (0.0)	0.7
Other	4	33.3 (28.2)	1.1 ± 16.5	0.0 (0.0)	0.8	0.2 ± 3.1	0.0 (0.0)	0.6

UHT, ultra-high temperature; SD, standard deviation; IQR, interquartile range; N, number.

**Table 3 nutrients-16-02104-t003:** Dairy iodine intake in different subgroups of the population.

Variables	Iodine Intake (μg)	Z	*p*
Gender			
Boys	28.1 (46.6)	−0.64	0.523
Girls	25.9 (43.6)		
Age			
<12 years	27.0 (44.6)	−1.00	0.315
≥12 years	25.4 (44.9)		
Province			
Liaoning	36.2 (40.2)	−8.47	<0.001
Yunnan	20.5 (39.0)		
Residence			
Urban	34.3 (39.8)	12.30	<0.001
Rural	19.1 (43.2)		
Ethnic			
Han	29.7 (46.5)	−3.70	<0.001
Other	23.1 (38.1)		
Total	26.8 (44.8)		

**Table 4 nutrients-16-02104-t004:** Selection of most parsimonious model based on selection criteria.

Models	AIC	BIC
Null model	19,866.81	19,864.81
Model I	19,831.89	19,829.89
Model II	19,666.51	19,664.51
Model III	19,650.54	19,648.54

NB: AIC = Akaike Information Criterion and BIC = Bayesian Information Criterion.

**Table 5 nutrients-16-02104-t005:** Generalized linear model for analysis of iodine intake in dairy products.

Variables	β	Standard Error	t	*p*
Intercept	14.221	2.545	5.59	
Gender	1.663	1.445	1.15	2.250
Age	4.770	2.152	2.22	0.027
Province	−9.448	1.524	6.20	<0.001
Residence	−17.958	1.457	−12.32	<0.001
Ethnic	−1.851	1.622	−1.14	0.254

**Table 6 nutrients-16-02104-t006:** Distribution of milk iodine intake and the proportion of the EAR, RNI, and UL for iodine.

	7~11 Years	12~14 Years
EAR (μg/d) [4]	65	80
RNI (μg/d) [4]	90	110
UL (μg/d) [4]	250	300
Distribution of children with iodine intake (%)		
<EAR	86.0	94.6
EAR~RNI	8.3	4.6
RNI~UL	5.6	0.8
≥UL	0.1	0.0
Iodine intake in this study (μg/d)	27.0	25.4
Proportion of EAR (%)	41.5	31.8
Proportion of RNI (%)	30.0	23.1
Iodine content per serving (μg)	57.5	57.5
Proportion of EAR per single serving (%)	88.5	71.9
Proportion of RNI per single serving (%)	63.9	52.3
Iodine content/300 mL (μg)	69.0	69.0
Proportion of EAR /300 mL (%)	106.2	86.3
Proportion of RNI /300 mL (%)	76.7	62.7

EAR, estimated average recommendation; RNI, recommended nutrient intake; UL, tolerable upper intake level.

## Data Availability

The data presented in this study are available on request from the corresponding author due to privacy.

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
