# Peer review of "Children’s Iodine Intake from Dairy Products and Related Factors: A Cross-Sectional Study in Two Provinces of China"

_nutrients, 2024, doi:10.3390/nu16132104_

Round 1
Reviewer 1 Report
Comments and Suggestions for Authors
Your manuscript triggers attention to the imp0ortance of dairy products as an important source of iodine. This is frequently neglected by colleagues working in the area of iodine deficiency as a public health problem in most countries in the world, except Europe, the USA, New Zealand, Australia, and similar countries. Therefore, your study is timely and important, and with data outside the mentioned regions where consumption of dairy products is high, and the animal feeds are enriched with iodine. Your manuscript is also very well written and clear, and the study, although no national, was well designed for the objectives that you proposed.
You have made your calculations based on the Recommended Nutrient Intake (RNI). I would like to suggest to also include the estimations based on the Estimated Average Recommendation (EAR) as this is the value used for predicting adequacy of the nutrients in populations. The deductions of your study are applicable to both individuals and populations, but for global application of the results the latter is more important. I would like to suggest to include calculations and deductions based on EAR values in the summary, as well as for section 3.5 (lines 205 to 215), and Table 5, and the corresponding paragraphs in the discussion section. Both EAR and RNI comparison are important in your manuscript. You should also mention that a food that provides more than 15% of the RNI is considered an excellent source of that nutrient, and which was the case for the dairy products in the two provinces of China that you studied.
In the discussion you included the comparison of the iodine content in the dairy products from China with those from other developed countries. However, you did not mention this detail in the summary of the manuscript. It would be sufficient to say in the summary that the content of iodine in the Chinese dairy products was intermediate to that reported in Europe and the USA.
Author Response
- Your manuscript triggers attention to the importance of dairy products as an important source of iodine. This is frequently neglected by colleagues working in the area of iodine deficiency as a public health problem in most countries in the world, except Europe, the USA, New Zealand, Australia, and similar countries. Therefore, your study is timely and important, and with data outside the mentioned regions where consumption of dairy products is high, and the animal feeds are enriched with iodine. Your manuscript is also very well written and clear, and the study, although no national, was well designed for the objectives that you proposed.
Response: Thank you very much for your affirmation. It makes us feel a great need to carry out this research.
- You have made your calculations based on the Recommended Nutrient Intake (RNI). I would like to suggest to also include the estimations based on the Estimated Average Recommendation (EAR) as this is the value used for predicting adequacy of the nutrients in populations. The deductions of your study are applicable to both individuals and populations, but for global application of the results the latter is more important. I would like to suggest to include calculations and deductions based on EAR values in the summary, as well as for section 3.5 (lines 205 to 215), and Table 5, and the corresponding paragraphs in the discussion section. Both EAR and RNI comparison are important in your manuscript. You should also mention that a food that provides more than 15% of the RNI is considered an excellent source of that nutrient, and which was the case for the dairy products in the two provinces of China that you studied.
Response: Thank you very much for your suggestion to add the EAR, which is a better way to evaluate the intake of a population than the RNI. We have added the calculations and deductions based on EAR values to the summary, result, table 5 and the corresponding discussion section. Foods with more than 20% of Daily Value are good sources of this nutrient according to nutrition labelling, and we have added these to the discussion section as well (line 21,240-259,277-279,311-319).
- In the discussion you included the comparison of the iodine content in the dairy products from China with those from other developed countries. However, you did not mention this detail in the summary of the manuscript. It would be sufficient to say in the summary that the content of iodine in the Chinese dairy products was intermediate to that reported in Europe and the USA.
Response: Thank you very much for your comment. We have added this sentence “the content of iodine was intermediate to that reported in Europe and the USA” to the summary and conclusion section (line 25-26,369-375).

Reviewer 2 Report
Comments and Suggestions for Authors
The study addresses a major and understudied issue in child nutrition. It reflects a well-designed and well-executed complex survey with randomization and good initial analysis using nonparametric methods owing to the reported difficulty meeting the usual assumptions for parametric least squares analysis.
However, the nonparametric initial analysis apparently does not extend to the multiple regression model estimation. Information is needed regarding the extent to which the usual regression assumptions are satisfied, to help establish model validity, and indications of model fit and predictive strength are needed.
The analysis is built around a complex sampling scheme, so the resulting data analysis needs to reflect the nested structure of the data selection process. This does not seem to be the case in the reported multiple regression results, so it is difficult to draw meaningful conclusions from the reported results if they do not take nesting into account.
The analysis is made more difficult by the fact that Table 1 shows very small group sizes for cases of milk intake greater than or equal to 300g/day for children in rural and with age over 12 years
The document provides some hints about possible implications of the findings. However, the broader impact of the results is likely to be enhanced greatly with additional discussion—maybe a couple of paragraphs—on implications for policy directions at national, provincial, and local levels and regarding how clinical practice might be impacted by the results.
Comments on the Quality of English Language
The document is quite readable, with only minor need for textual revision.
Author Response
- The study addresses a major and understudied issue in child nutrition. It reflects a well-designed and well-executed complex survey with randomization and good initial analysis using nonparametric methods owing to the reported difficulty meeting the usual assumptions for parametric least squares analysis.
Response: We are grateful for your positive feedback on our study addressing a significant issue in child nutrition.
- However, the nonparametric initial analysis apparently does not extend to the multiple regression model estimation. Information is needed regarding the extent to which the usual regression assumptions are satisfied, to help establish model validity, and indications of model fit and predictive strength are needed.
Response: Thank you for your insightful comments. We reconsidered the data and decided to use a generalized linear mixed model for the multivariate analysis, the model building process and model selection have been added to the article (line 147-158,219-228).
- The analysis is built around a complex sampling scheme, so the resulting data analysis needs to reflect the nested structure of the data selection process. This does not seem to be the case in the reported multiple regression results, so it is difficult to draw meaningful conclusions from the reported results if they do not take nesting into account.
Response:Thank you for your insightful comment regarding the complex sampling scheme and its implications for the data analysis. We agree that it is essential to account for the nested structure of our data in the analysis to draw meaningful conclusions. We decided to use a generalized linear mixed model (GLLMM)to solve this problem.GLMMs, by integrating the capabilities of GLMs and LMMs and introducing random effects, are capable of dealing with more complex data structures, especially datasets with hierarchical or nested characteristics, such as multi-center studies, or naturally stratified data, GLMMs can consider the correlation between data points, thereby providing more accurate model estimates.
- The analysis is made more difficult by the fact that Table 1 shows very small group sizes for cases of milk intake greater than or equal to 300g/day for children in rural and with age over 12 years
Response: I agree with your opinion. The number of children in this study who achieved a daily intake of more than 300g was small, especially in rural areas, there were only 11 children, and these 11 children were all younger than 12 years old, and there were no rural children older than 12 years old who consumed more than 300g of milk per day. So we can only use a chi-square test to determine whether the proportion of urban and rural children achieving a dairy intake of more than 300g per day is the same, and whether the proportion of younger and older children achieving a dairy intake of more than 300g per day is the same. The results showed that the proportion of older children who drank milk up to standard was significantly lower than that of younger children, and that rural children were significantly less likely to drink milk than urban children, suggesting that we need to pay attention to the insufficient intake of dairy products by older children in rural areas.
- The document provides some hints about possible implications of the findings. However, the broader impact of the results is likely to be enhanced greatly with additional discussion—maybe a couple of paragraphs—on implications for policy directions at national, provincial, and local levels and regarding how clinical practice might be impacted by the results.
Response: Thank you for your insightful comments and suggestions. We agree that discussing the broader impact of our findings in terms of policy directions at national, provincial, and local levels, as well as their implications for clinical practice, would greatly enhance the value of our work.We have carefully considered your feedback and have added the paragraph to the discussion section of our manuscript(line 336-354).
Round 2
Reviewer 2 Report
Comments and Suggestions for Authors
The revisions are totally on-topic, making an already strong manuscript considerably more powerful.
Author Response
Response:Thank you very much for your comments and affirmations, I've learned a lot of new things in the process of revising.